# A System Dynamics Simulation Applied to Healthcare: A Systematic Review

**DOI:** 10.3390/ijerph17165741

**Published:** 2020-08-08

**Authors:** Mohammad Reza Davahli, Waldemar Karwowski, Redha Taiar

**Affiliations:** 1Department of Industrial Engineering and Management Systems, University of Central Florida, Orlando, FL 32816, USA; wkar@ucf.edu; 2Université de Reims Champagne-Ardenne, 51100 Reims, France; redha.taiar@univ-reims.fr

**Keywords:** simulation modeling, system dynamics, healthcare

## Abstract

In recent years, there has been significant interest in developing system dynamics simulation models to analyze complex healthcare problems. However, there is a lack of studies seeking to summarize the available papers in healthcare and present evidence on the effectiveness of system dynamics simulation in this area. The present paper draws on a systematic selection of published literature from 2000 to 2019, in order to form a comprehensive view of current applications of system dynamics methodology that address complex healthcare issues. The results indicate that the application of system dynamics has attracted significant attention from healthcare researchers since 2013. To date, articles on system dynamics have focused on a variety of healthcare topics. The most popular research areas among the reviewed papers included the topics of patient flow, obesity, workforce demand, and HIV/AIDS. Finally, the quality of the included papers was assessed based on a proposed ranking system, and ways to improve the system dynamics models’ quality were discussed.

## 1. Introduction

Complex and intractable healthcare problems, as well as limited progress in discovering effective solutions, continue to challenge the healthcare industry [1,2]. In complex healthcare systems, the mutual effects of different variables are subtle, while the impact of interventions over time is not immediately identifiable [3,4,5,6,7]. Furthermore, many internal and external factors that influence healthcare outcomes are often nonlinear [3,4,5,6,7]. Under these conditions, simulation models can help to elucidate the counterintuitive behavior of complex healthcare problems [6]. Simulation modeling is considered to be a wise option because the complexities of problems are far beyond our capacity to solve them manually [6]. According to Sterman [6], it is challenging to develop accurate healthcare simulation models because there is no reliable and well-understood principle in this area. However, the substantial benefits of developing and applying simulation models are learning about complex problems and testing different interventions [6]. 

Operations research (OR) approaches, especially simulation methods, have been used to specify complex problems in healthcare since the 1960s [8]. Many healthcare aspects, such as extended care, specialty care, public health, hospital operation, rehabilitation, and long-term care, have been subject to such computer simulations in the past [9]. Simulation methods can be categorized into four main groups: Monte Carlo (MC), discrete-event simulation (DES), system dynamics (SD), and agent-based simulation (ABS) [10]. Each type of simulation has its advantages concerning specific aspects of healthcare [10]. For example, DES has been commonly used in medical emergency system applications, while SD was found useful in epidemic modeling and disease prevention efforts [10,11]. Other factors may also affect the applicability of the simulation method, including the goals, significance of the feedback loop, simulation time, amount and quality of the input data, and type of healthcare units [10,12]. Generally speaking, matching the most suitable simulation method with a specific aspect of healthcare is not an easy task [10]. 

System dynamics (SD) simulation emerged in late 1950 as a result of focusing on the behavior of complex systems over a specific period [11]. The main features of SD simulation are a feedback loop, stocks and flow function, and time delay [11]. These features have been used to model the emerging nonlinearity of systems behavior [11]. In SD, dynamic objects move though flows and accumulate in the stocks [13]. At each specified time, stocks calculate and report the quantitative status of the objects [13]. Feedback loops and delays in SD can create a dynamic system [13]. Therefore, if the change occurs anywhere in the system, it will result in a chain reaction throughout the system [13]. Furthermore, SD studies can be carried out qualitatively or quantitatively [10]. For example, if the objective of the study is to explain the problem in a better way or reliable quantitative information is not available, the study can be conducted qualitatively [10]. 

In general, the SD model can be developed through specific steps, including a clear explanation of the problem, generating a qualitative diagram of the system structure, converting the qualitative hypothesis to a quantitated simulation model, testing the model, and informing policy decisions about the model’s implications [14]. The main advantages of SD simulation are discovering the emergent properties and characteristics of a system, creating quantitative analysis for qualitative problems, identifying the most important system parameters, predicting the long-term effects of decisions, and helping stakeholders learn about the nature of their problems [15]. The main disadvantages of SD simulation are the inability to model low-level interactions and overreliance on qualitative validation [15]. 

The main objective of the present study is to review recent papers that utilize the SD method to analyze a variety of healthcare problems. We attempt to provide an overview of SD’s applications in healthcare by discussing different modeling approaches, procedures, and quality. This review is structured as follows: The first part presents the main outcomes of published review papers in healthcare. The Methodology section discusses the inclusion and exclusion criteria and the risk of bias. The Results section provides the outputs of the literature search, while the Discussion section explains healthcare aspects and research areas of studies. Finally, the quality of the reviewed papers is evaluated in the Quality Assessment section.

## 2. Evaluation of Review Papers 

To date, there have been several literature reviews regarding the applications of simulation methods in healthcare. Some of the highly cited literature reviews published between 2000 and 2019 are shown in Table 1. 

Two methodologies were used to summarize the published papers of interest: (a) the literature review, and (b) the systematic literature review. Of the 19 papers considered here, six papers used the literature review method, while 13 papers used the systematic literature review method. Of the systematic review papers, only He and associates [21] used the Preferred Reporting Items for Systematic Reviews and Meta-Analyses (PRISMA) guidelines for reporting a systematic review. The majority of the papers review multiple simulation methods, and almost half of the papers concern the overall healthcare area. The average and median numbers of included papers among these reviews were 270 and 182, respectively. More specifically, the maximum number of papers used in the published reviews was 1441 [17], while the minimum number of papers was six [23]. 

Web of Science was the most popular database among the selected review papers, followed by PubMed, Science Direct, and Google Scholar. The outcomes of the review papers can be classified into five distinct groups, including (a) the categorization of the included papers based on country, publication source, and year; (b) grouping by areas of research; (c) comparing simulation techniques; (d) discussing trends of topics or trends of simulation techniques; and (e) describing specific areas of research in detail.

Two major sections in the selected review papers are suggestions for research and study limitations. In the suggestions portions of the papers, many of the review papers offer interesting topics for future research, while others attempt to describe their upcoming studies. Additionally, a few review papers suggest novel methodologies for conducting a literature review. The limitations sections in the review papers can be categorized into different groups. The first limitation is identifying and selecting papers published in a specific time frame. The second limitation is the inability to identify individual relevant records resulting from the use of a restricted number of search keywords. The third limitation is not covering all search databases, while the fourth limitation is restricting the search to specific regions or countries.

## 3. Methodology

The present systematic literature review followed PRISMA guidelines [30] and contains two main features: developing research questions and determining the search strategy. The following research questions were formulated for this review:
**RQ1**.What are the main problems in the healthcare sector that have been studied in the past using the SD approach?**RQ2**.How was the SD approach utilized to model and address complex healthcare problems?**RQ3**.What can be learned from the past SD simulation research in the healthcare area that will support high-quality research in the future?


To answer the above research questions, a search strategy was developed to identify and review the relevant scientific papers. The search strategy included (a) defining keywords and identifying the relevant materials, (b) filtering the articles written by two authors, and (c) addressing the risk of bias [30]. As mentioned earlier, one of the main concerns in a systematic review is developing comprehensive and accurate keywords. The main objective of the present systematic review was to target all aspects of healthcare, such as medicine, management, costs, systems, technology, safety, and system operations. Since this is a broad and multi-faceted topic, it is important to generate unique keywords that cover all aspects of healthcare, including health, medical care, healthcare, and the healthcare system. Since classical medical research focuses on disease, health is defined as the absence of disease [31]. Referring to the definition and metrics of health described by the World Health Organization (WHO) and the global burden of disease (GBD), the main health-related words are diseases, healthy, life expectancy, morbidity, aging, and illness [31]. 

Health care, healthcare, or health-care is the improvement or maintenance of health via the diagnosis, prevention, and treatment of illness, disease, injury, and other mental and physical impairments in individuals [32]. Physicians, health professions, primary care, secondary care, tertiary care, and public health are part of healthcare [32]. Medical care is defined as the professional treatment of disease, illness, or injury [33]. A health care system, health system, or healthcare system is an institution or organization of individuals and resources that deliver health-related services to respond to the needs of target populations [34]. In conclusion, we developed the first set of keywords based on the presented definitions, as shown in Table 2. 

Web of Science and Google Scholar were used as database search tools in the present review. The search conducted in this review was divided into two steps: (a) the first set of keywords was used to discover relevant articles (1162 articles have been identified and added to the central database), and (b) a new set of keywords was developed based on the identified papers, which was used to discover additional articles. The second set of keywords is shown in Table 3. The second set of keywords was developed to identify papers that did not mention specific keywords—including health, medical, disease, or illness—but whose contents are related to our review. Finally, 1231 articles with relevant content were identified.

After the development of the main database and the identification of all relevant papers, a formal screening process was applied to the database based on the exclusion and inclusion criteria. The inclusion criteria were as follows:

Articles with SD models or an SD procedure/framework;
Articles related to healthcare;Articles published from 2000 to 2019;Articles related to research questions;Articles written in English.


The exclusion criteria were as follows:
Papers written in other languages;Chapters of books;Review papers;Articles from secondary sources that were not free or open access;Letters, newspaper articles, viewpoints, presentations, anecdotes, duplicated studies, short papers, and posters.


In this systematic review, bias could occur by (a) applying inclusion/exclusion criteria and (b) specifying the dimensions of the included papers. To address the first type of bias, two researchers (authors) independently reviewed the titles, abstracts, and conclusions of the identified records to choose articles for full-text review. Subsequently, the two authors compared their selected articles to create a unified list. After reading the full texts of the selected articles, the authors decided whether or not to include the articles. If there was an agreement between the two researchers, the article was considered and included. Disagreements between the two authors were resolved in sessions with the third author. To address the second type of bias, two authors independently specified aspects of healthcare and modeling in the included papers then compared the results, resolving disagreements by consulting with the third author. 

## 4. Results

A chart of the selection strategy according to PRISMA guidelines is displayed in Figure 1. As a result, the present review included a total of 253 papers written by 940 authors. On average, a single paper was written by four authors. As shown in Figure 2, the trend of papers included increased over time. 

Interestingly, only 15% of the included papers were published between 2000 and 2009, and there was a significant increase in the number of included papers after 2013. To confirm these results, we first compared the trend of included papers with that of unscreened records, as shown in Figure 3.

Secondly, we compared our results with those of other literature reviews. Kunc and associates [16] indicate that there was a significant and increasing trend in SD publications after 2010 in the areas of health improvement, health services, health epidemics, and health policy-making, although the trends were significantly more dramatic than those observed in our results. Torres [17] presented the number of papers cited in the System Dynamics Review Journal and found that only 37% of the citations were from 2000 to 2009. 

Of the 253 included papers, 207 were published in journals and 46 were presented as conference papers. The most popular sources of publication among the included papers, in order, were the Journal of the Operational Research Society, System Dynamics Review, Winter Simulation Conference, PLoS One, Systems Research and Behavioral Science, and Health Care Management Science.

## 5. Discussion 

The papers included in the present review are categorized into different aspects of healthcare, as shown in Table 4. The most popular categories among the included papers are healthcare operations, communicable diseases, non-communicable diseases, and healthcare systems. The most popular subcategories among the included papers are patient flows, HIV/AIDS, obesity, and workforce demand. 

### 5.1. Patient Flow

The most popular research area among the reviewed papers is patient flow. Since healthcare facilities are under pressure to decrease delays and reduce costs, an investigation into streamlining patient flow is beneficial [35]. The objectives of the included papers in this area can be classified into analyzing the length of stay, studying the shift, delayed discharge, and developing SD models. Patient flow was analyzed in the context of admission, emergency departments, cardiac catheterization services, elderly patients, and hospitals in general. For SD model simulation, information held by hospital databases and staff was used for model calibration. The most common simulation time for patient flow modeling was one day to 2 weeks, and the most common simulation scenarios were changes in the number of beds, the value of demand, and the number of hospital staff. Almost half of the reviewed papers in this area used case studies, and the majority of the papers were quantitative studies.

It has been reported that more than 75% of instances of delayed discharge were a result of delays in making long-term care arrangements for elderly patients [36,37]. Several studies evaluated the impact of hospital resources on the length of stay [35,38,39], while others focused on elderly individuals, chronically ill patients, and their impact on bed occupancy, the average length of stay, and resource consumption [40,41]. The main conflicting results are the impact of changing bed capacity on performance. A study by Kumar [42] indicates that optimizing bed capacity could minimize patient waiting time in an urban country hospital. Lane and associates [39] conclude that reducing the number of beds does not increase the waiting time for emergency admissions. At the same time, Taylor and Dangerfield [43] mention that increasing bed capacity is not a practical solution for improving the average waiting time in cardiac catheterization services. Moreover, Rashwan and associates [37] report that increasing bed capacity in post-acute care services has a temporary impact and is insignificant in the long term. A study by German and associates [44] evaluated healthcare services in the Philippines and indicates that hospital bed requirements could be satisfied by 2021. Several studies analyzed decision-making in healthcare and blamed (a) applying informal policies, (b) focusing on short-term results, and (c) using a single-performance measure for problems in the process of decision-making [43,45].

The key variables of the simulation models include the patients awaiting discharge, inpatients waiting for rehabilitation, patients in care, length of stay, admission rate, bed capacity and demand, and waiting lists. The main output variables are the average daily occupancy, patient waiting time, number of patients, number of delayed discharges, and post-acute accessibility. The core part of the stock-flow diagram among the included papers for the patient flow model is illustrated in Figure 4.

In order to develop an expanded SD model, auxiliary variables such as the average daily arrivals, referrals, and walk-in admissions can be linked to the patient admission/arrival rate flow. Different auxiliary variables can be linked to patients in hospital or patients on hospital wards such as the numbers of occupied beds, patients, and empty beds. A variety of auxiliary variables can be connected to, and have an impact on, the patient treatment rate such as the treatment time, treatment conditions, and size of hospital staff. The patient in assessment for discharge can be linked to different auxiliary variables, including the inpatient discharge, the patient needs for other treatment, the patient needs convalescence, and the patient needs rehabilitation. The core part of the causal loop diagram for the reviewed papers is shown in Figure 5.

### 5.2. Obesity 

Another popular research area in healthcare is obesity. Increasing rates of obesity and the incidence of being overweight are putting pressure on the economy and health services [46]. Many diseases and conditions, such as type 2 diabetes mellitus, coronary heart disease, cancer, hypertension, and musculoskeletal disorders, are associated with obesity [46]. The included papers investigate obesity among children [47,48,49,50], women of reproductive age [51], individuals in low-income urban areas [52], and lower-socioeconomic status (SES) groups [53,54]. The objectives of the selected papers can be categorized into two main groups: (a) reducing obesity (reactive responses) and (b) presenting interventions for obesity prevention (proactive responses).

Papers report different results for reducing and preventing obesity. For example, Abdel-Hamid [46] indicates that combining physical activity and diet composition is critical for effective obesity treatment; Abidin and associates [55] emphasized that efficient eating behavior to prevent obesity is a combination of decreasing food portions, controlling meal frequency, and limiting the intake of high-calorie foods. Furthermore, Roberts and associates [50] suggested implementing a comprehensive combination of different interventions to decrease obesity, including increasing the availability of healthy food, obesity prevention programs, bans on junk food advertising, and subsidies for healthy food. Allender and associates [56] discussed the causes of obesity in four domains: fast food and junk food, social influences, physical activity, and participation in sports. Although a study places more emphasis on building outdoor activity spaces for children [48], it was also reported that physical education is more effective than physical activity in reducing obesity among children [49]. Interestingly, Chen and associates [54] concluded that enhancing an individuals’ status from lower- to middle-income levels would be effective in controlling obesity and being overweight.

The most popular scenarios among the reviewed papers are changing the sensitivity of weight to diet, healthy food availability, weight loss rates, the availability of educational programs, and preventing fast food advertising. The key variables among the reviewed papers include the average daily energy intake, the expected weight of a user, change in expected weight, energy expenditure, energy metabolism, body composition, the effect of diet on weight, the sensitivity of weight to exercise, overweight adults, obese adults, normal-weight children, overweight children, obese children, physical activity, changes in average weight per year, participation in sports, junk food consumption, levels of physical activity, healthy weight, healthy food choices, and mental health. The main output variables of the reviewed papers in this area include weight, change in weight, carbohydrate intake and oxidation, fat intake and oxidation, change in physical activity, the number of overweight and obese individuals, and the prevalence of being overweight and obesity. The results indicate that qualitative studies, building causal loop diagrams, and integrating SD with other simulation methods are not common in obesity papers. The core part of the stock-flow diagram among the included papers for the obesity model is shown in Figure 6.

To expand the core model, different auxiliary variables can be linked to the weight loss rate and weight gain rate, including changes in average weight per day/month, engaging in weight loss/gain behaviors, daily energy balance, and healthy weight.

### 5.3. HIV/AIDS and Tuberculosis

Studying the application of SD in HIV/AIDS and tuberculosis (TB) is popular among researchers. In many countries, AIDS is the main cause of death in the adult population and a significant source of excessive demands on existing treatment and medical resources [57]. With the high cost of HIV/AIDS and the absence of medical solutions, behavioral approaches and prevention programs are the main solutions for preventing the spread of the epidemic and decreasing individual exposure to HIV [57]. The objectives of these studies can be categorized into different groups: (a) analyzing the impacts of different therapies and programs on diseases, (b) studying HIV care services, and (c) studying effective interventions.

Some of the main programs are mentioned in the following paragraph. Several studies have investigated the consequences of highly active antiretroviral therapy (HAART) [58,59,60]. HAART is the most suitable treatment for advanced HIV disease in developed countries and works by preventing the virus from replicating [58]. Other studies have investigated multidrug-resistant tuberculosis (MDRTB) control programs, HIV harm-reduction programs, and evidence-based HIV intervention programs [57,61].

Interestingly, almost all the papers mention model calibration, and different types of data are used for this purpose (e.g., European Network for HIV/AIDS Surveillance (ECDC) data [58], and data from literature reviews, the Estonian Ministry of Social Affairs, and the National Institute for Health Development [61]). The average simulation time among the papers is 10.5 years. The scenarios of the selected papers include changing therapy cure rates, populations sharing needles (PSH), the drug injection frequency (DIF), the preventive-program population coverage, the percentage of the target population with easy access to the program, the proportions of motivated and unmotivated members, the length of the programs, the proportions of non-drug users and active drug users, implementing new needle-exchange programs, financial hardship, program budgets, health education, and psychological counseling.

While Dangerfield and associates [58] indicate that implementing HAART could decrease new HIV infections, a study by Lebcir and associates [60] states that the decrease in HIV- and TB-related deaths would be small for HAART coverage of up to 50% of individuals, and that coverage of 70% or greater would be more effective. Lounsbury and associates [62] highlight that access to antiretroviral therapy (ART) would improve outcomes for HIV patients, while other studies report that upon the implementation of effective harm-reduction programs, the number of HIV-associated deaths would decrease by 30% [61]. Furthermore, Atun and associates [61] stated that effective harm-reduction programs would more effectively decrease the cumulative deaths from TB than effective multidrug-resistant tuberculosis (MDRTB) control programs. Lebcir and associates [60] did not mention effective harm-reduction programs, concluding that a high MDRTB cure rate has decreased the deaths from TB. Miller and associates [57] noted that shortening the evidence-based HIV prevention program would increase the number of individuals finishing the program. Zou and associates [63] indicate that the most efficient scenarios that decreased HIV infection involved condom promotion, needle-exchange programs, and psychological counseling. Lastly, a study by Shariatpanahi and associates [64] on disease awareness distribution indicated that disease awareness distribution for AIDS and tuberculosis is much faster than for breast cancer and autism.

The key variables in the developed models include the program participants, undiagnosed HIV-positive individuals with AIDS, family awareness, the support of family and friends, undiagnosed HIV, and diagnosed HIV. The main output variables include the annual HIV incidence, cumulative TB deaths, cumulative deaths from AIDS, cumulative MDRTB deaths, cumulative deaths from AIDS in non-TB individuals, and HIV infections and incidences.

### 5.4. Workforce Demand

One of the main research areas of healthcare is workforce demand. The objectives of the selected papers on workforce demand are planning and predicting the number of workforces, including medical specialists [65,66], the pediatric workforce [67], the cardiac surgery workforce [68], dentists [69], the ophthalmologist workforce [70], the radiology specialization workforce [71], physicians [72,73], registered nurses [74], and physical therapists [75]. Interestingly, the average simulation time among the selected papers in this area is 24.5 years, which is higher than that for previous research areas of healthcare. The main scenarios among these papers are changing the level of population growth, the medical school graduation rate, the national exam pass rate, medical demand, an aging population profile, the physician-to-population ratio, the level of student intake, and the age of retirement.

A study by Barber and López-Valcárcel [65] indicates that the shortage of medical specialists in Spain would increase from 2% in 2010 to 14.3% in 2025 with moderate population growth. Wu and associates [67] mention that the predicted decline in the supply of Taiwanese pediatricians could be delayed by interventions, while Vanderby and associates [68] model the future Canadian cardiac surgery workforce with different scenarios and a predict short-term excess of cardiac surgeons by 2030. Samah and associates [69] state that, according to the simulation, the total available Malaysian clinical dentists in the year 2030 will be fewer than the required dentists, while a study conducted about Singapore [70] reports that in all scenarios, the number of Singaporeans with eye conditions will double and thus there will be a significant increase in eyecare demand and a great need for ophthalmologists. Taba and associates [71] indicate that the Australian system will never meet the demand for radiologists in the simulation time. Ishikawa et al. [73] state that by 2020, the physician shortage in the Hokkaido Prefecture region of Japan will be resolved, and De Silva and associates [72] highlight that the shortage of doctors in Sri Lanka will be resolved by 2025. Lastly, Morii and associates [75] project that the number of physical therapists in Japan will be 1.74 times greater in 2025 and 2.54 times greater in 2040 than in 2014.

The critical variables among the models include immigration, emigration, the targeted workforce, the general population, births, deaths, graduated students and medical specialization schools, demand based on the population, and retirement. The main output variables among these papers include the number of specialists, population prediction, workforce demand and supply prediction, care supply, and demand. The core part of the stock-flow diagram for the included papers for workforce demand is shown in Figure 7.

Auxiliary variables can be linked to flows, including the application rate, immigrant workforce, national exam pass rate, graduation rate, national exam application rate, graduate school enrolment rate, rate of choosing a hospital as a workplace, and employment rate.

### 5.5. Chronic Diseases

The main area of non-communicable diseases is chronic diseases, defined as *conditions that last a year or more and require ongoing medical attention and/or limit activities of daily living* [76]. It has been reported that the use of the term “chronic disease” has large variation [77]. For example, the Centers for Disease Control (CDC) uses the term “chronic diseases” for cancer, type 2 diabetes, coronary heart disease, stroke, obesity, and arthritis; however, the Centers for Medicare and Medicaid Services include further conditions such as Alzheimer’s disease and depression [77]. It is reported that the same kind of variation in terminology exists in academic papers [77]. The authors of the reviewed papers take different paths to address chronic diseases, where some only focus on one area of chronic diseases—such as diabetes, cancer, or cardiovascular disease—while others address all aspects of chronic diseases. The objectives of this group of papers can be divided into different sets (1) projecting growth and the number of individuals with chronic diseases, (2) developing a simulation model of chronic disease, and (3) analyzing interventions, therapies, treatments, and programs to improve chronic care.

The average simulation time among the selected papers is 32.4 years, although a study by de Andrade and associates [78] used 1440 min as the simulation time in an SD model. The main scenarios of the included papers are changes in the intervals and intensity of treatments and therapies, stage-specific survival rates, patients and population size, adopting different levels of programs and interventions, and the existence of other conditions such as obesity.

A study by Tejada and associates [79] indicates that the best screening policy for breast cancer prevention is annual breast cancer screening for all women aged 65–80 years old, while Liew [80] emphasizes that either clinical breast examination (CBE) or mammographic screening has the potential to avoid some deaths from breast cancer. A study by Palma and associates [81] highlights that prostate-specific antigen (PSA) screening would be beneficial in decreasing prostate cancer mortality, while Hallberg and associates [82] mention important points for developing an SD model for patients with growing tumors, which were (a) categorizing patients based on a physiological point of view, and (b) considering a non-constant rate for tumor growth.

Kenealy and associates [83] concluded that implementing anti-smoking interventions has the potential to decrease cardiovascular events, while Lich and associates [84] identified a broad hypertension control effort to be the most effective intervention in increasing the quality and length of life (quality-adjusted life-years). De Andrade and associates [78] discussed the cause of delays in the treatment of ST-Elevation Myocardial Infarction (STEMI) patients, the main factors of which can be categorized as professional, equipment, and transportation logistics. A study by Recio and associates [85] investigates the impact of exposure to urban noise on the growth of cardiovascular mortality. Additionally, the improvement of chronic diseases has been investigated by implementing different interventions, such as increasing access to primary care [86], evaluating the behavior of cardiovascular drugs on the market [87], increasing income and employment, enhancing social cohesion and neighborhood attractiveness [88], and improving physician education and care coordination [89]. Mehrjerdi [90] analyzed the interconnections between health problems and weight, and found that teaching individuals about their health will have a considerable impact on the number of deaths from heart attacks.

Several studies indicate that the rates of diagnosed diabetes and prediabetes can be decreased by different interventions such as improving the clinical management of diabetes, reducing obesity prevalence, and innovative healthcare delivery, while other studies project that the population with diabetes will increase in the U.S. and Japan [91,92].

The key variables among the reviewed papers are access to treatment/therapies, age, adherence to treatment/therapies, the rate of chronic disease diagnoses, chronic disease mortality, moderate and severe stages of chronic disease, the population in different age groups, population growth, and the population with disease. The main output variables among the reviewed papers include the treatment/therapy cost, number of disease incidents and diagnoses, number of deaths, and fraction of the population with the disease or those at-risk. The core part of the stock-flow diagram for chronic diseases is displayed in Figure 8.

Some studies implement the entire core part of the stock-flow diagram [84,89,91,93,94,95], while other studies implemented only the diagnosed part (blue box), in which there is an additional flow from the population to diagnosed patients [82,96,97,98]. Furthermore, several of the selected studies only implemented the undiagnosed-to-diagnosed part of the stock-flow diagram (orange box) without expanding on different stages of chronic diseases [79,80,81,83,85,87,92,99]. Depending on the chronic disease, different auxiliary variables can be linked to the onset and progression flows, including the smoking prevalence, the use of primary care, stress, poor diet, physical activity, the transition rate between the different stages of chronic disease, the management of chronic disease, the elderly portion, and the obese portion. Screening, testing, and detection variables can be linked to the diagnosis flow.

### 5.6. Hybrid Modeling

One of the main research areas in healthcare is hybrid modeling and the comparison of simulation methods. The main objective of the selected papers is to discuss the integration or comparison of SD with other simulation methods in a healthcare context. Interestingly, all papers are qualitative studies, with the majority being conference papers published at the Winter Simulation Conference. The main results of the reviewed papers include (a) the advantages of using hybrid models [100,101,102], (b) selecting an appropriate simulation technique for different healthcare problems [103,104,105], and (c) presenting a framework for integrating simulation methods [106,107,108,109,110,111,112].

### 5.7. Chlamydia

Chlamydia—one of the research areas of communicable diseases—was studied by five papers [113,114,115,116,117]. The main objectives of the selected papers were to analyze interventions and screening programs. Interestingly, the simulation time in four of the five papers was 1 year, and all were quantitative studies. Several studies analyzed the potential cost–benefit of chlamydia screening programs [113,114,115,117]. The key stock variables include the different stages of the disease, susceptibility to sequelae, transmission incidents, and recovery. The main output variables among the papers are the cases of infertility prevented per year, cases of pelvic inflammatory disease (PID) prevented per year, number of sequelae vs. the screening rate, infection prevalence, screening rate, and per-capita cumulative cost. The core part of the stock-flow diagram for the reviewed papers for chlamydia is shown in Figure 9. It is possible to expand the core stock-flow diagram to cover multiple stages of the disease and low- and high-risk groups.

### 5.8. Other Research Areas

A handful of papers present SD models as the main results and outcome, including for a methadone maintenance treatment [118], water regulation in the human body [119], depression [120,121], fatigue [122], alcohol misuse [123,124], urban community health services [125], construction workers’ health behavior [126], Hispanic immigrant worker health [127], patient satisfaction [128,129], urban health policy-making [130], healthy eating [131], oral health [132], the total costs associated with hip joint endoprosthesis [133], mobile stroke units [134], occupational health and safety (OHS) performance [135,136], and electronic health information exchange (HIE) [137,138].

## 6. Quality Assessment

To build guidelines for SD applications in healthcare, we targeted (a) the categorization and analysis of relevant papers and (b) an assessment of the quality of SD models. The assessment criteria were adopted from a previous study since they provide an analytical and quantitative approach to analyzing the reviewed papers [3]. However, since those criteria were developed for quantitative studies with simulation models, the present criteria were adjusted to cover (a) qualitative and quantitative studies and (b) papers with and without SD models. The list of the quality criteria is shown in Table 5.

A good quality paper is one that has clear objectives and results, provides detailed information about developing the SD model, presents model variables, and, in the case of quantitative studies, describes scenarios, model validation, outcome variables, and the source of the input data. The quality of the papers is categorized into four groups: “good”, “medium”, “low”, and “very low”. The results of this quality assessment are shown in Figure 10.

It should be noted that more than 50% of the reviewed papers belong to the good- and medium-quality categories. Furthermore, the observed trend in quality has not increased over the 20 years. In fact, in the last five years, there has been a considerable number of papers classified as “bad” and “very bad” quality. Generally, papers belonging to the aspects of communicable diseases, non-communicable diseases, and disorders and stress exhibited higher quality than those belonging to other healthcare aspects, which may indicate evidence-based practice in these areas. It is important to note that we only assessed the information given in the papers and their additional/attached files. Moreover, we found that some highly cited papers did not present their models, and therefore, we adjusted the criteria to assess the papers based on model description or model presentation and not only based on model presentation.

The clarity of the study objectives is vital to modeling efforts and is related to the type of problems under consideration [6]. The reviewed papers’ objectives can be divided into different groups, such as developing models, analyzing elements/problems in a system, improving a system with different factors, and projecting elements in a system.

The quality of data was assessed based on two parameters: (a) the engagement of experts and stakeholders in terms of the survey, interview, discussion, community-based model development, and group model building, and (b) the presentation of a source of data for model calibration. A total of 65 out of the 253 papers engaged experts and stakeholders in their studies or used interviews or surveys in their research.

The reviewed papers have been categorized into different groups: presenting stock-flow diagrams, presenting causal loop diagrams, presenting SD frameworks, and describing SD model development. A total of 26 out of the 253 papers present no SD model, 11 of these 26 papers describe different frameworks for hybrid modeling, and the remaining 15 papers describe model development. The reviewed papers present 139 causal loop diagrams and 249 stock-flow diagrams. A total of 32 out of the 253 papers only present the causal loop diagram, 74 present both the casual loop diagram and the stock-flow diagram, and 121 only present the stock-flow diagram.

The time horizon is one of the main aspects of stock-flow diagram development and should go back far enough in history to show the emergence of problems and their consequences and symptoms. The time horizon must also extend far enough into the future to elucidate the effects of different policies and delays [6]. The choice of time horizon strongly affects the perception and assessment of problems and the evaluation of policies [6]. The included papers use different time horizons in their research, from a half of a day [139] to 120 years (1920–2040) [85], as shown in Figure 11.

The model validation can be a problematic task and should, therefore, be performed by engaging stakeholders and experts during the model-building process, where the stakeholders should understand and trust the behavior and internal structure of the model [40]. Relying too much on qualitative validation is one of the weaknesses of SD [140]. Hence, we divided model validation among the selected papers into three groups:
Direct structural tests, including empirical and theoretical tests;Structure-oriented behavioral tests, such as extreme condition tests and sensitivity tests;Behavioral pattern tests, considered to be model calibration [140].


It should be noted that out of the 235 papers selected for this review, 187 studies were carried out quantitatively. A total of 115 out of these 187 papers discussed calibration tests by comparing the model results with different sources, including (a) actual/historical/experimental data, and/or (b) data extracted from the literature reviews, and/or (c) data from surveys. A total of 57 reported direct structural tests, including empirical tests and theoretical tests. Finally, 104 studies performed sensitivity tests by evaluating the response of the model to fluctuation in the values of certain parameters.

A total of 187 quantitative papers contain a total of 707 and an average of four outcome variables in the forms of charts, tables, and graphs. The cost, number, and percentage of specific parameters over time are the most common forms of the outcome variable. The most popular outcome variables of the reviewed papers are the number of patients, individuals, or doctors; the cost of the treatment, disease, or program; and the mortality or diagnosis rate.

The included papers presented a total of 410 scenarios. The main types of scenarios were changing the value of established variables, adding new feedback loops or variables to the model, and removing specific variables. Common scenarios among the reviewed papers include program, legislation, or guideline adaptation; running the model with and without a specific system or feedback loop; changing the percentage or size of the population or population growth; changing the level of health services; changing hospital performance indicators; changing the rate of birth, death, or disease; and changing the treatment procedure or cost. The included papers used a total of 8627 variables in their models and an average of 22 variables per model. Moreover, papers included 2835 equations in their models and an average of 11 equations per stock-flow diagram.

Typically, SD simulation modeling has been used to study and explain high-level problems in complex healthcare systems, mainly related to the impact of strategy and policy decisions on health outcomes, and to design future healthcare systems [6]. The main advantages of SD applications in the healthcare sector include the relatively low cost of collecting input data, transforming complex problems into simple structures, and quickly constructing suitable models [6,141]. However, one of the important limitations of the SD modeling approach is that it cannot provide a deep description of the micro-behaviors of healthcare systems. Another disadvantage is the difficulty of validating complex SD models with a large number of inter-related variables [6,141].

Finally, to generate a better picture of the included papers, the titles and abstracts were also analyzed. Figure 12 and Figure 13 show a map of co-occurrence in two forms: a bubble chart and a heatmap. In Figure 12, the nodes correspond with specific terms, and their sizes indicate the frequency of term occurrence. Links between two nodes represent the co-occurrence of the terms in publications. Frequently co-occurring terms form clusters and emerge closer to each other with the same color. A first glance at Figure 12 reveals the central cluster (red color) with terms including system dynamics, approach, study, strategy, framework, technique, insight, and hybrid simulation. The next cluster (green color) identifies critical issues such as community, organization, public health, implementation, policy intervention, collaboration, and prevention. In Figure 13, darker colors represent a higher frequency of the analyzed terms.

## 7. Conclusions

The current paper offers a comprehensive review of the critical healthcare issues under consideration by the SD application community. A carefully chosen set of keywords was used for discovering relevant articles. Following PRISMA guidelines, we explored three main research questions related to fundamental problems in the healthcare sector that have been studied in the past using the SD approach. These questions focused on how the SD methodology has been utilized in the past and what could be learned from the application of the SD simulation approach. Finally, the specific criteria for inclusion were defined and applied to the subsequent analysis focusing on developing useful SD applications in the healthcare sector. In general, reviewing and assessing contributions related to SD applications in the healthcare sector is somewhat tricky because there is no unique or standardized way of developing and presenting such papers. However, the main criterion applied here was the overall organization of the manuscript and the quality of the technical content from which the reader could learn about the current trends in applying the SD approach to healthcare problems.

The research questions were answered as follows:
**RQ1**.What are the main problems in the healthcare sector that have been studied in the past using the SD approach? To answer this question, the included records have been reviewed, and the main problems were extracted and are represented in Table 4.**RQ2**.How was the SD approach utilized to model and address complex healthcare problems? To answer this question, essential aspects of the SD approach, including “core part of SD diagrams”, “key variables”, and “output variables” are represented in the Discussion section.**RQ3**.What can be learned from the past SD simulation research in the healthcare area that will support high-quality research in the future? In order to answer this question, the main aspects of high-quality SD research have been assessed in the quality assessment section, such as the clarity of the objectives and results, the description of the scenarios, the source of the input data, and model validation.


This paper contributes to the body of knowledge related to developing system dynamics and simulation models that have been used in the past to analyze complex healthcare systems. We draw on a systematic literature selection and present a comprehensive view of system dynamics’ applications to the healthcare sector. The study includes papers from 2000 to 2019 and demonstrates the broad range of interests in the research community and the variety of healthcare topics. We demonstrate that the most popular research topics include patient flow, obesity, workforce demand, and HIV/AIDS. Moreover, the quality of the reviewed papers has been assessed using a proposed ranking system. Finally, the need for improving the quality of the system dynamics models was discussed.

## Figures and Tables

**Figure 1 ijerph-17-05741-f001:**
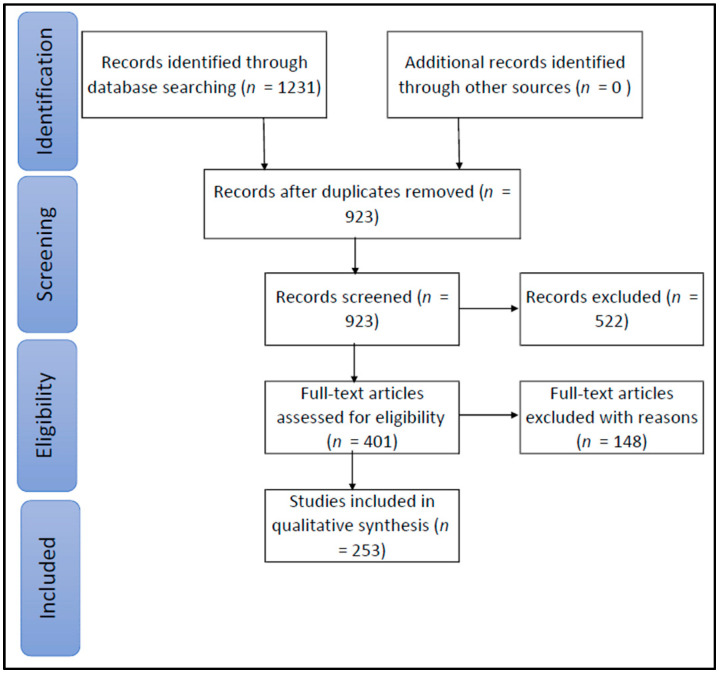
Chart of the selection strategy.

**Figure 2 ijerph-17-05741-f002:**
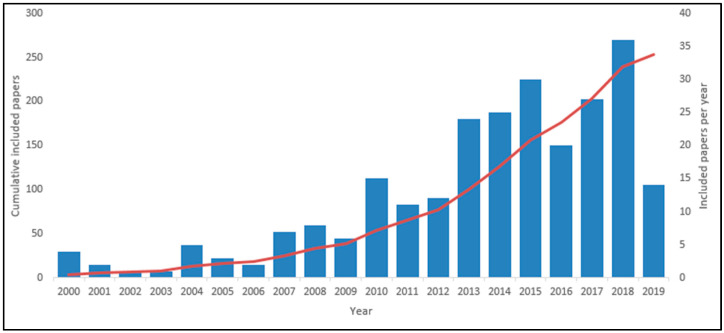
Included papers per year.

**Figure 3 ijerph-17-05741-f003:**
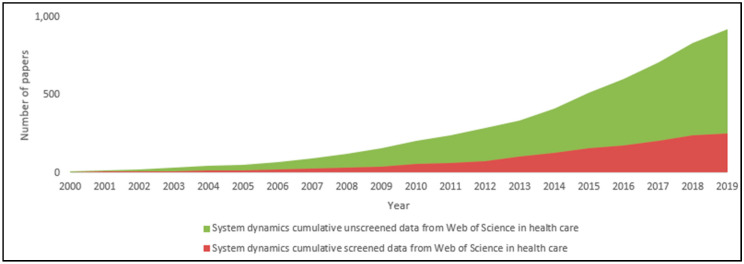
Published papers before and after the screening process.

**Figure 4 ijerph-17-05741-f004:**
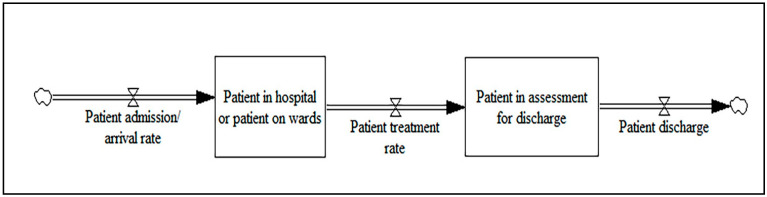
The core part of the stock-flow diagram for the patient flow model.

**Figure 5 ijerph-17-05741-f005:**
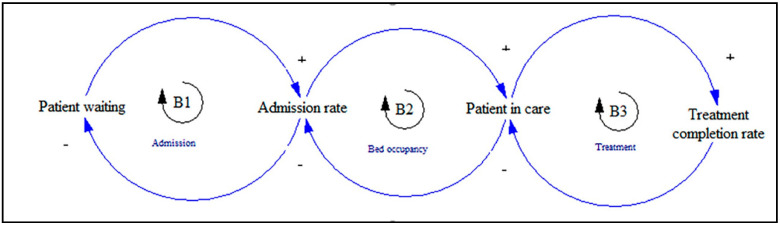
The core part of the causal loop diagram for patient flow.

**Figure 6 ijerph-17-05741-f006:**
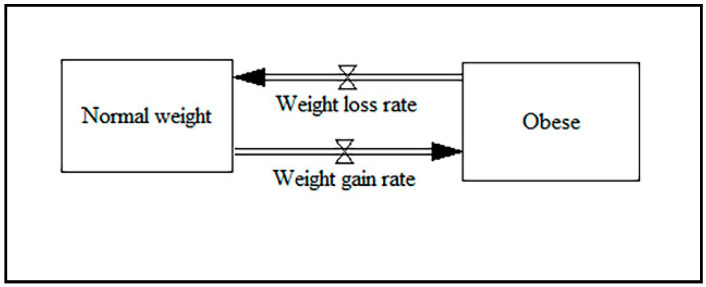
The core part of the stock-flow diagram for obesity.

**Figure 7 ijerph-17-05741-f007:**
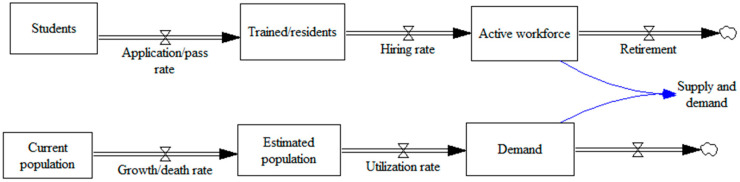
The core part of the stock-flow diagram for workforce demand.

**Figure 8 ijerph-17-05741-f008:**
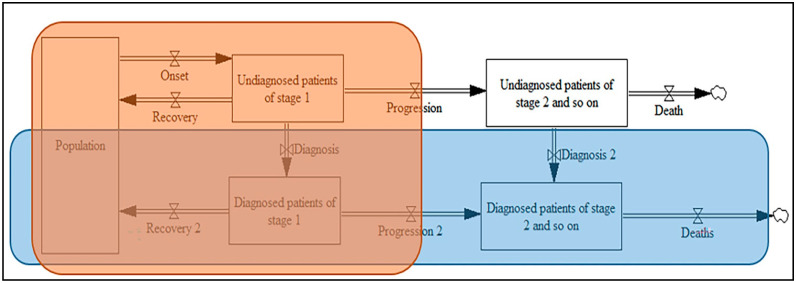
The core part of the stock-flow diagram for chronic diseases.

**Figure 9 ijerph-17-05741-f009:**
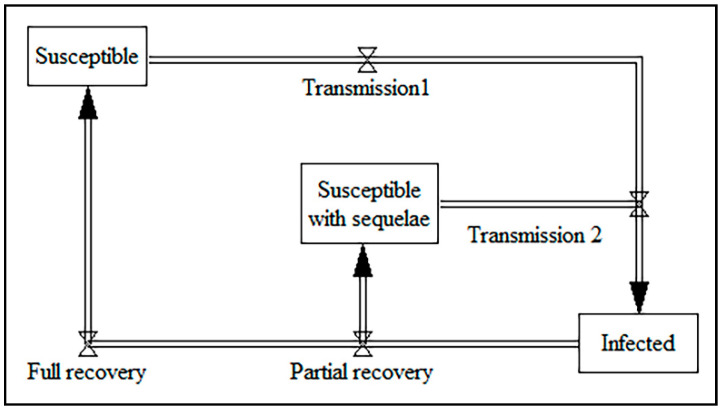
The core part of the stock-flow diagram for chlamydia.

**Figure 10 ijerph-17-05741-f010:**
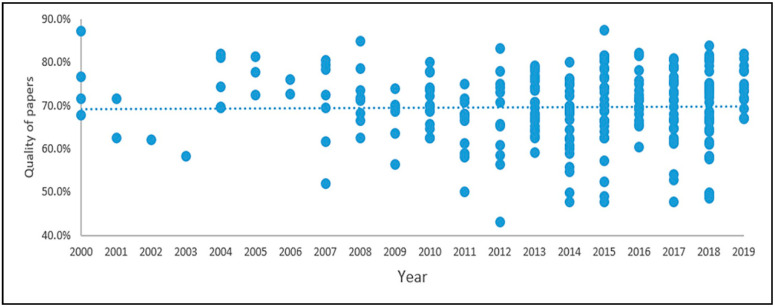
Quality assessment of the reviewed papers (legend: good quality > 80%, 80% > medium quality > 70%, 70% > bad quality > 65%, 65% > very bad quality).

**Figure 11 ijerph-17-05741-f011:**
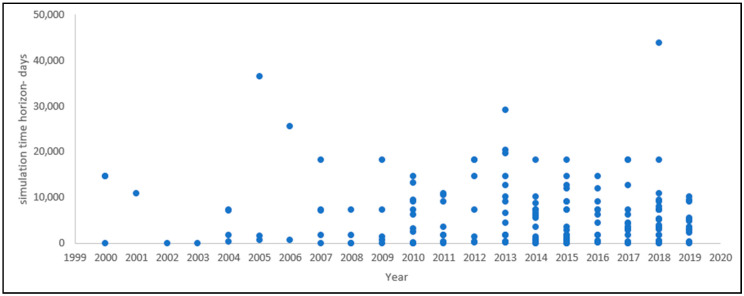
Time horizons among the selected papers.

**Figure 12 ijerph-17-05741-f012:**
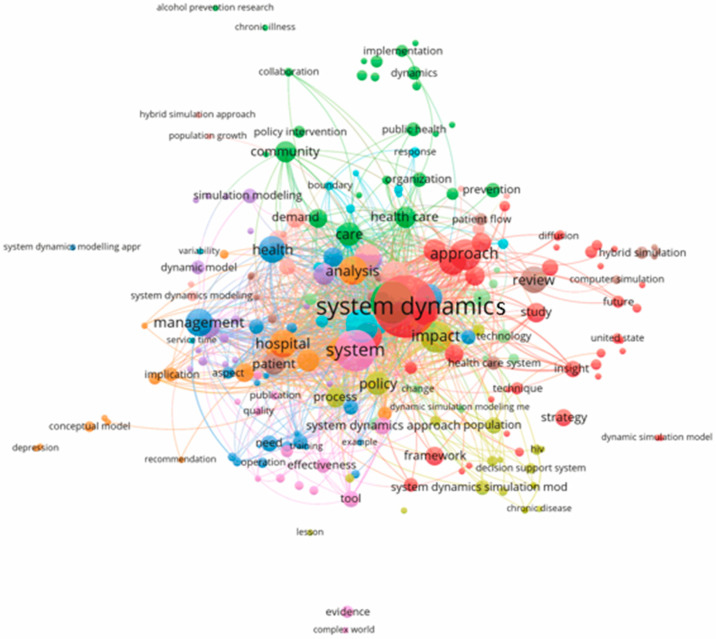
The map of the co-occurrence of terms in titles and abstracts of included papers.

**Figure 13 ijerph-17-05741-f013:**
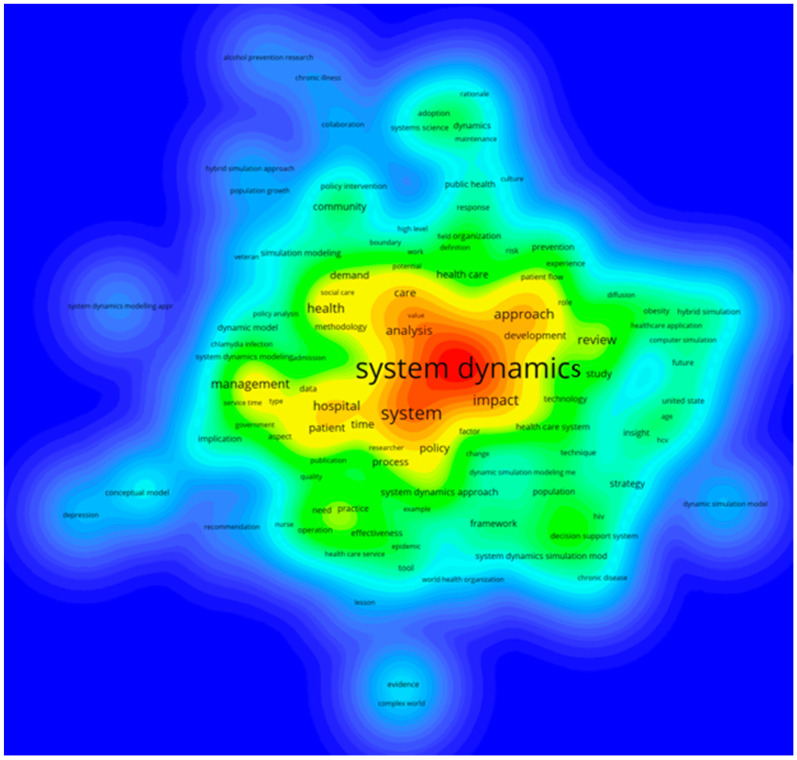
The heatmap in terms of titles and abstracts of included papers.

**Table 1 ijerph-17-05741-t001:** Review articles related to simulation modeling in healthcare and published from 2000 to 2019.

Reference	Area of Application	Simulation Method	Number of Papers Included
[1]	Healthcare	Operations research approaches	342
[2]	Healthcare	Different simulation methods	251
[3]	Healthcare delivery and population health	Simulation techniques	182
[11]	Healthcare	Different simulation methods	201
[13]	Healthcare	Different simulation methods	232
[16]	All fields	System dynamics	800
[17]	All fields	System dynamics	1441
[18]	Emergency departments	Different simulation methods	106
[19]	Emergency medical services (EMS)	Simulation models	24
[20]	Emergency departments	Different simulation methods	254
[21]	Inpatient bed management	Modeling techniques	92
[22]	Healthcare	Discrete-event simulation	211
[23]	Health policy	System dynamics	6
[24]	Healthcare	Discrete-event simulation	Not mentioned
[25]	Healthcare	Simulation methodologies	Not applicable
[26]	Rheumatoid arthritis	Modeling techniques	58
[27]	Cancer care	Operations research approaches	90
[28]	UK healthcare	Operations research approaches	142
[29]	Mental health	Different simulation methods	160

**Table 2 ijerph-17-05741-t002:** The first set of keywords used in the present review.

Row	Set
Test set 1	(Health OR health care OR medical care)
Test set 2	(System dynamics OR SD)
Search 1	#1 AND #2
Test set 3	(Disease OR illness OR treatment OR injury OR morbidity)
Search 2	#3 AND #2

**Table 3 ijerph-17-05741-t003:** The second set of keywords used in the present review.

Row	Set
Test set 1	(System dynamics OR SD)
Test set 2	(Disorder OR stress)
Test set 3	(Drug OR medication)
Test set 4	(Human body OR human behavior)
Search 3	#1 AND (#2 OR #3 OR #4)

**Table 4 ijerph-17-05741-t004:** Classification of the papers included in the present review.

Category	Subcategory	Number of Included Papers
Aging and population	Healthcare demand	3
	Evolution of population	1
Communicable diseases	HIV/AIDS, tuberculosis	11
	Chlamydia	5
	Hepatitis C	2
	Influenza	2
	Ebola virus	1
	Foot-and-mouth disease	1
	Middle East Respiratory Syndrome (MERS) Coronavirus (CoV)	1
	Severe acute respiratory syndrome coronavirus (SARS-CoV)	1
	Other infectious diseases	11
Cost and price in healthcare	Pharmaceutical fees	1
	Cost–benefit analysis	1
	Insurance economics	1
	Patient dispensing fees	1
	Performance-based payment system	1
Disorder and stress	Obesity	14
	Depression	3
	Body Water Homeostasis	1
	Fatigue	1
	Post-traumatic stress disorder (PTSD)	1
	Workplace stress	1
Drugs and medications	Alcohol misuse	3
	Opioid use and misuse	2
	Nicotine product	1
	Pain medicine	1
	Use/abuse of drugs	1
Healthcare operations	Patient flow	17
	Emergency department	7
	Hospital	3
	Patient service centers	2
	Behavior of nurses	1
	Clinical knowledge	1
	Clinical workforce	1
	Neonatal care services	1
	Outpatient clinics	1
	Patient access to general practice	1
	Radiotherapy department	1
	Referrals system	1
	Telehealth	1
Healthcare systems	System improvement	4
	Mental health care	2
	Urban health	2
	Community health services	2
	Construction workers’ health	1
	Curative and preventive services	1
	Healthcare delivery systems	1
	Healthcare affordability and accessibility	1
	Immigrant worker health	1
	Military psychological health system	1
	National healthcare system	1
	One Health concept	1
	Public health	1
	Rural minority health	1
	Sustainability in healthcare	1
	Sustainable health care	1
	Youth health	1
	Other healthcare systems	4
Human body and behavior	Body weight	1
	Functional loss	1
	Healthy eating	1
	Lower back pain (LBP)	1
	Oral health care	2
Medical treatment and devices	Chemotherapy	1
	Childhood immunization	1
	Disease awareness	1
	Eyecare services	1
	Hip joint endoprosthesis	1
	Home hemodialysis (BASIC-HHD)	1
	Knee implants	1
	Maintenance procedure	1
	Mobile stroke units	1
	Prospective Health Technology Assessment (ProHTA)	1
Non-communicable diseases	Diabetes	7
	Cancer	4
	Cardiovascular disease	4
	Dementia	3
	Intellectual disabilities (ID)	2
	Asthma	1
	Cognitive impairment	1
	End-stage renal disease	1
	Hernia	1
	Other chronic diseases	7
Healthcare process and policy	Health interventions	5
	Long-term care policy	2
	Care planning	1
	Clinical decision thresholds	1
	Lean deployment strategies	1
	Policy implementation	1
	Strategic planning	1
	Successful health programs	1
	Workforce planning	1
Safety in healthcare	Occupational safety and health	2
	Risk of adverse events	1
	System safety	1
Simulation in healthcare	Simulations comparison	7
	Hybrid modeling	6
	Introduction to system dynamics	1
Specialty in neonatology, obstetrics, gynecology	Infant mortality	4
	Cesarean delivery	1
Supply and demand in healthcare	Workforce demand	12
	Supply chain	3
	Care supply and demand	1
	Blood supply chain	1
	Internal service supply chains	1
	Medical demand	1
	Need for facilities	1
Technology and information in healthcare	Electronic health records	4
	Health information exchange (HIE)	3
	Cybersecurity	1
	Health information technology	1
	Health information systems	1
	Sharing of information	1

**Table 5 ijerph-17-05741-t005:** List of quality criteria.

Row	Quality Criteria
1	Presenting clear objectives
2	Presenting clear scenarios and interventions
3	Presenting clear outcomes variables by graphs, charts, or tables
4	Describing the development of an SD model/framework or presenting a detailed model/framework
5	Presenting and explaining model parameters
6	Improving the quality of data by using stakeholders’ engagement, surveys, interviews, and databases
7	Validating models
8	Presenting clear results

Legend: Score is from 0 (not mentioned) to 2 (fully described). Rows 1, 4, 5, 6, and 8 applied qualitative studies. Rows 1 to 8 applied quantitative studies.

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
