# Peer review of "A System Dynamics Simulation Applied to Healthcare: A Systematic Review"

_ijerph, 2020, doi:10.3390/ijerph17165741_

Round 1
Reviewer 1 Report
In this review article, the authors reviewed system dynamics simulation models in the analysis of complex healthcare problems. This article is well written and informative. The introduction is logical and the rationale is reasonable. The methodology is clear and can be followed. Results and discussion are well described and can be accepted in the current version. The conclusion is made based on the evidence. Congratulation on this nice review.
Author Response
Thank you for these comments.
Reviewer 2 Report
Thanks for the opportunity to review this paper. An important contribution to the field.
Author Response
Thank you for these comments.
Reviewer 3 Report
ijerph-857986
This is a clear and very well written synthesis of the systems dynamic modelling/simulation literature in relation to health care interventions, policy and planning. It provides an in-depth oversight of the literature and will be a useful reference paper for health service researchers, policy makers, planners and managers.
Author Response
Thank you for these comments.
Reviewer 4 Report
In this paper, the authors proposed to form a comprehensive view for applying system dynamics to healthcare problems and did a systematic review of relevant papers from 2000 to 2019. The results provide an in-depth overview of the system dynamics simulation methods that have been developed and utilized to solve or explain complex problems in the area of contemporary healthcare. The reviewer believes readers/researchers from the healthcare field intended to do system dynamics studies will benefit greatly from this paper.
p.3 line 111-115
In this paper, the authors present a systematic literature review following 3 formulated research questions RQ1-3. RQ1 and 2 seem to be answered in Section 5. Discussion. But the answer for RQ3 doesn't to be explained in this paper. Section 6. Quality assessment seems to be the answer, but not as clear as that of RQ1-2. It might be very helpful for the readers if the authors can add a small paragraph to explain where those answers for RQ1-3 be found in the paper.
p.18 line 526-545
On p.3 Figure1. the authors identified a total of 253 papers from the selection process. But in this section, the authors discussed insights among 187 papers, which the authors did not explain how those 187 papers were selected out of the 253 papers.
Author Response
Thank you for these comments.
We have clarified these points as follows.
Comment: p.3 line 111-115
Revision: p.20 line 593-602
The research questions were answered as follows:
RQ1. What are the main problems in the healthcare sector that have been studied in the past using the SD approach? To answer this question, the included records have been reviewed, and the main problems were extracted and represented in Table 4.
RQ2. How was the SD approach utilized to model and address complex healthcare problems? To answer this question, essential aspects of the SD approach, including “core part of SD diagrams,” “key variables,” and “output variables” were represented in the discussion section.
RQ3. What can be learned from the past SD simulation research in the healthcare area that will support high-quality research in the future? In order to answer this question, the main aspects of high-quality SD research have been assessed in the quality assessment section, such as clarity of objectives and results, description of the scenarios, the source of input data, and model validation.
We have clarified this point….
Comment:p.18 line 526-545
Revision: p.17 line 531-532
It should be noted that out of the 235 papers selected for this review, 187 papers were carried out quantitatively.
Reviewer 5 Report
Review
A system dynamics simulation applied to healthcare: A systematic review
This paper contributes to the literature appealing to the interest in developing system dynamics and simulation models to analyze complex healthcare systems. It draws on a systematic literature selection and presents a comprehensive view of the applications of system dynamics to healthcare. The study includes papers from papers from 2000 to 2019 and demonstrates the latest interest of the research community and the variety of healthcare topics. It was shown that the most popular research areas among included papers are patient flow, obesity, workforce demand, and HIV/AIDS. Moreover, the quality of the papers included in the study was assessed using a proposed ranking system. Finally, conclusions and discussion is provided concerning the improvement of the quality of the system dynamics models.
In general, reviewing and criticizing this kind of work is rather difficult because there is no unique way to write such papers. However, there are points, which concern mainly the overall organization of the manuscript, which the reader expects to see and learn from this endeavor.
The present paper meets these standards and offers the reader a comprehensive review of the issues under investigation. It followed PRISMA guidelines and by developing research questions and determining the search strategy. Research questions such as what are the main problems in the healthcare sector that have been studied in the past using the SD approach, how the SD approach has been utilized, and what has been learn from the application of SD simulation research, serve as core dimensions for the analysis and the presentation of reviewed papers. The Web of Science and Google Scholar were used as database and search tool. A set of keywords was set for discovering relevant articles while based on those identified papers, a following set of keywords was applied. From them inclusion criteria were shaped and applied to the subsequent analysis and presentation of the results.
The result section is well presented and it is important that the authors comment on pros and cons of the methods used, so it becomes apparent what we have learnt from system dynamical research of this type.
In conclusion, I consider this work a valuable contribution to the literature and I recommend publication in this form.
Author Response
We have added a short discussion related to the benefits of SD applications in healthcare as follows:
Revision: p.17 line 552-559
Typically, SD simulation modeling has been used to study and explain high-level problems in complex healthcare systems, mainly related to the impact of strategy and policy decisions on health outcomes, and to design future healthcare systems [6]. The main advantages of SD applications in the healthcare sector include the relatively low cost of collecting input data, transforming complex problems into simple structures, and quickly constructing suitable models [6,141]. However, one of the important limitations of the SD modeling approach is that it cannot provide a deep description of the micro behaviors of healthcare systems. Another disadvantage is the difficulty of validating complex SD models with a large number of inter-related variables [6,141].
In addition, we have revised the Conclusion section as well to read as follows:
Revision: p.19-20 line 579-612
The current paper offers a comprehensive review of the critical health care issues under consideration by the SD application community. A carefully chosen set of keywords was used for discovering relevant articles. Following PRISMA guidelines, we explored three main research questions related to fundamental problems in the healthcare sector that have been studied in the past using the SD approach. These questions focused on how the SD methodology has been utilized in the past, and what could be learned from the application of the SD simulation approach. Finally, the specific criteria for inclusion were defined and applied to the subsequent analysis focusing on developing useful SD applications in the healthcare sector. In general, reviewing and assessing contributions related to SD applications in the healthcare sector is somewhat tricky because there is no unique or standardized way to develop and present such papers. However, the main criterion applied here was the overall organization of the manuscript and the quality of technical content from which the reader could learn about the current trends in applying the SD approach to healthcare problems.
The research questions were answered as follows:
RQ1. What are the main problems in the healthcare sector that have been studied in the past using the SD approach? To answer this question, the included records have been reviewed, and the main problems were extracted and represented in Table 4.
RQ2. How was the SD approach utilized to model and address complex healthcare problems? To answer this question, essential aspects of the SD approach, including “core part of SD diagrams,” “key variables,” and “output variables” were represented in the discussion section.
RQ3. What can be learned from the past SD simulation research in the healthcare area that will support high-quality research in the future? In order to answer this question, the main aspects of high-quality SD research have been assessed in the quality assessment section, such as clarity of objectives and results, description of the scenarios, the source of input data, and model validation.
This paper contributes to the body of knowledge related to developing system dynamics and simulation models that have been used in the past to analyze complex healthcare systems. We draw on a systematic literature selection and present a comprehensive view of system dynamics' applications to the healthcare sector. The study includes papers from 2000 to 2019 and demonstrates the broad range of interests in the research community and the variety of healthcare topics. We demonstrate that the most popular research topics include patient flow, obesity, workforce demand, and HIV/AIDS. Moreover, the quality of the reviewed papers has been assessed using a proposed ranking system. Finally, conclusion and discussion are provided concerning the need for improving the quality of the system dynamics models.